# The Psychological Impact of the COVID-19 Lockdown: A Comparison between Caregivers of Autistic and Non-Autistic Individuals in Italy

**DOI:** 10.3390/brainsci12010116

**Published:** 2022-01-15

**Authors:** Laura Fusar-Poli, Miriam Martinez, Teresa Surace, Valeria Meo, Federica Patania, Chiara Avanzato, Maria Salvina Signorelli, Eugenio Aguglia

**Affiliations:** Psychiatry Unit, Department of Clinical and Experimental Medicine, University of Catania, Via Santa Sofia 78, 95123 Catania, Italy; miriam.martinez@outlook.it (M.M.); teresa.surace70@gmail.com (T.S.); valeriameo26291@gmail.com (V.M.); pataniafederica@gmail.com (F.P.); chi4493b@gmail.com (C.A.); maria.signorelli@unict.it (M.S.S.); eugenio.aguglia@unict.it (E.A.)

**Keywords:** autism, coronavirus, family caregiver, distress, general health, resilience

## Abstract

The COVID-19 outbreak has disrupted the daily routine of the population worldwide, including autistic people and their caregivers, with severe consequences on mental health. On one hand, the reduced social contacts and the interruption of outpatient and daycare services during the lockdown have represented a real challenge for autistic people and their caregivers. On the other hand, confinement has allowed individuals to spend more time pursuing their interests and stay home with their family members without feeling the pressure of social expectations. The present study aimed to compare the levels of personal wellbeing, family distress, insomnia, and resilience between caregivers of autistic people and caregivers of people with other neurodevelopmental, psychiatric, or relational disabilities. A web survey was completed by 383 participants, of which 141 were primary caregivers of autistic people. We did not find any significant difference between caregivers of autistic and non-autistic people in any of the considered psychological variables. Lower age of the autistic family member and lower resilience levels were significantly associated with higher individual distress in the group of caregivers of autistic people. Our findings do not corroborate the hypothesis that caregivers of autistic individuals have had more severe consequences than other caregivers during the lockdown. However, they confirm the importance of promoting resilient coping strategies in autistic people and their caregivers.

## 1. Introduction

Autism is a life-long neurodevelopmental condition that, according to the Diagnostic and Statistical Manual for Mental Disorders, Fifth Edition (DSM-5), is characterized by difficulties in social communication and the presence of restricted patterns of interests or behaviors [1]. Characteristics of autism are widely variable in their presentation and can sometimes be masked due to learned coping or camouflaging strategies [2,3]. For this reason, the medical model of autism has proposed three levels of severity depending on the support needed by the autistic individual [1].

Along with core features, autistic people might present with irritability, challenging behaviors [4], and self-injury [5], especially in the case of associated intellectual disability (ID) [6], which is present in at least one-third of this population [7]. Conversely, individuals with average or above-average cognitive abilities are more frequently affected by psychiatric co-occurring conditions, such as depression, anxiety, attention deficit-hyperactivity disorder (ADHD), or sleep problems [8,9]. Medical morbidities, such as epilepsy, metabolic disorders, or pain-related conditions, are also highly prevalent among the autistic population [10,11,12]. Accessibility to specialized healthcare for medical investigations and developmental services for skills improvement, autonomy support, and quality of life is therefore needed [6,13].

Depending on the severity of the condition, autistic people, as well as individuals with other neurodevelopmental, psychiatric, or relational disabilities, may need to be supported by a caregiver during the activities of daily living. A caregiver is “a person who takes care of someone not self-sufficient and needs of global and continuous care, that can be the spouse, the other part of the civil union or cohabiting partner, a family member, or a relative” (Legge n. 205, 2017). The scientific community has highlighted how this figure may represent a category at higher risk of developing physical and mental health issues than the general population [14]. Particularly, it has been reported that being the caregiver of an autistic person—typically one of the parents—can be stressful and impact family functioning [15]. The level of perceived distress can vary according to the severity of the condition, the presence of psychiatric and medical co-occurring conditions, the intensive interventions needed, and the difficulties encountered to obtain targeted services [16]. Professional guidance is therefore needed to improve the knowledge about the autistic condition and gain emotional and relational support [17].

During the first wave of the COVID-19 pandemic, all non-essential healthcare and social care services were forcedly stopped. Although it was recommended to replace medical outpatient visits with remote contacts [18,19], in Italy many services were found unprepared to implement e-health facilities [20]. The interruption of healthcare services has caused severe consequences for people with disabilities, including autistic individuals. First, many rehabilitation activities dedicated to autistic people (e.g., daycare centers) were interrupted during the lockdown [21]. Second, changing daily habits has represented a challenge for autistic people, who may require a rigid adhesion to routines. Learning new habits like hand washing, use of masks, social distancing, and restricted outdoor access may have not been fully comprehended by autistic people causing heightened anxiety, frustration, and emotional breakdowns [22,23,24]. In this regard, the Italian legislation has contemplated some exceptions to the rigid rules imposed on the general population. In fact, individuals with disabilities, including autistic people, were allowed to circulate without protective masks and perform outdoor activities if needed.

Autistic people and their families have been diversely impacted by confinement. According to a recent systematic review, parents and caregivers of autistic individuals reported an overall increase in perceived stress and a decrease in psychological wellbeing during the COVID-19 outbreak [25]. However, findings regarding the psychological distress of the autistic family members were contradictory, depending on variables such as the age of the autistic subject, the severity of the condition, as well as the type of family structure with their daily living habits [25]. Focusing specifically on the Italian situation, Levante et al. showed that families of autistic children reported greater behavioral problems and parental distress than families of typically developing children during the lockdown [26]. An online survey also revealed that managing changes and restrictions during the lockdown has been very challenging for parents of autistic children. The majority of parents reported an increase in the burden of care compared to the pre-pandemic period [27]. Despite the enormous disruptions caused by the lockdown, experiences were not always described as straightforwardly negative in the literature. Some autistic adults reported, for instance, more flexibility, time, and space to pursue their own interests and passions as well as to stay with their families [28]. Others described that the confinement period has positively impacted their mental health, due to the less perceived societal expectations and the opportunity to behave in a more spontaneous way without spending too much effort camouflaging behaviors [29]. A study conducted in Spain showed that autistic adults perceived a decrease in stress levels after the lockdown onset and found an improvement in their psychopathological status after two months of social distancing; conversely, caregivers of both autistic children and adults reported higher stress levels [30].

In a previous paper, we showed that family caregivers of people with disabilities, including autism, reported more severe individual and family distress compared to the general population during the lockdown due to COVID-19. Moreover, we found higher levels of insomnia and lower resilience levels in the caregiver population [31]. To our knowledge, no studies have compared the psychological and familial wellbeing of caregivers of autistic people to those of caregivers of people with other neurodevelopmental or psychiatric disorders. Thus, with the present study we aimed:
(1)To compare the levels of psychological wellbeing, family distress, insomnia, and resilience perceived by the caregivers of autistic people during the lockdown in Italy to those perceived by the caregivers of people with other neurodevelopmental disorders and psychiatric or relational disabilities.(2)To evaluate factors associated with the levels of individual distress reported by caregivers of autistic people during the lockdown.


## 2. Materials and Methods

### 2.1. Study Design

Detailed information regarding study design and data collection has been reported in our previous paper [31]. A web-based questionnaire was developed by the research team and spread through social networks (i.e., Facebook) between 19 April and 3 May 2020. Moreover, associations dedicated to the support of people with mental and relational disorders and their caregivers were contacted to help spread the questionnaire (see Acknowledgments section).

In the present paper, we included only caregivers older than 18 years who were living with an autistic family member, or with family members having other neurodevelopmental disorders (e.g., ID, ADHD) or psychiatric disorders during the lockdown period. Caregivers of people with dementia and physical or neurological disorders were excluded from the present analysis. 

The questionnaire investigated demographic variables, characteristics of the family member with mental or relational impairments, and psychological outcomes. Outcome variables were assessed using the General Health Questionnaire-12 items (GHQ-12) [32,33], the Insomnia Severity Index (ISI) [34], the Brief Resilient Coping Scale (BRCS) [35,36], and the Family Distress Index (FDI) [37]. The Activities of Daily Living (ADL) scale [38] was used to evaluate the level of independence of autistic and non-autistic family members. Online consent was obtained from all participants. The study was conducted according to the Declaration of Helsinki and was approved by the University of Catania Psychiatry Unit review board. 

### 2.2. Measures 

#### 2.2.1. General Health Questionnaire-12 Items (GHQ-12) 

The GHQ-12 [32,33] is a brief, simple, easy-to-complete measure to screen for general (non-psychotic) psychiatric morbidity. Its application across different cultures and research settings as a screening tool is well-documented. The scale asks whether the respondent has experienced a particular symptom or behavior over the last two weeks. Each item is rated on a four-point scale (less than usual, no more than usual, rather more than usual, much more than usual). The most common scoring methods are the bi-modal (0-0-1-1) and the Likert scoring method (0-1-2-3). In our study, the GHQ-12 was scored according to the Likert scoring method, as suggested by Banks et al. [39]. Thus, the total score could range from 0 to 36, with higher scores indicating worse conditions. We considered a cutoff of 12/13 as the optimal threshold for psychological distress [40]. The Italian version of the GHQ-12 proved to be a reliable instrument, as indicated by a Cronbach’s alpha of 0.8 [32]. In our sample, Cronbach’s alpha was 0.74, indicating acceptable internal consistency.

#### 2.2.2. Family Distress Index (FDI) 

The FDI is an 8-item questionnaire developed to obtain self-report observations regarding the occurrence of family hardships, as well as challenges that reflect family disharmony and family intolerance [37]. The FDI is easily scored by summing individual items which are rated on a 4-point Likert scale (0–3), with total scores ranging from 0 to 24. Scores higher than 7 reflect greater cohabiting distress. We asked the participants to focus on the distress perceived during the past month. According to Fonseca et al. [41], scores higher than 12 may indicate the presence of significant family distress. The FDI has not been formally validated in Italian but was translated by the authors. In our sample, the FDI had a Cronbach’s alpha of 0.83, indicating good internal consistency.

#### 2.2.3. Insomnia Severity Index (ISI) 

The ISI has been designed to be both a brief screening measure of insomnia and an outcome measure for use in treatment research [34]. The scale content corresponds in part to DSM-IV criteria for insomnia and measures the subject’s perception of symptom severity, distress, and daytime impairment over the past weeks. The total score can range between 0 and 28, and the scores can be interpreted as follows: no clinically significant insomnia (0–7); subthreshold insomnia (8–14); clinical insomnia—moderate severity (15–21); clinical insomnia—severe (22–28). According to the validation study, the Italian version of the ISI has an internal reliability coefficient of 0.75 [34]. In our study, we calculated a Cronbach’s alpha of 0.90, indicating excellent internal consistency.

#### 2.2.4. Brief Resilient Coping Scale (BRCS) 

The BRCS [35,36] is a 4-item tool designed to measure basic coping skills and resilience. It explores the areas of coping skills, control overreactions, growing from difficulties, and resilience. The scoring method is a 4-point Likert scale, and total scores can range between 4 and 20. According to the scores, respondents can be classified into low resilient copers (4–13), medium resilient copers (14–16), and high resilient copers (17–20). The BRCS has not been validated in the Italian language. In our sample, the BRCS had a Cronbach’s alpha of 0.80, indicating good internal consistency. 

#### 2.2.5. Activities of Daily Living (ADL) 

Activities of daily living (ADL; [38]) is a term used in healthcare to refer to people’s daily self-care activities. Basic ADL includes bathing and showering, personal hygiene and grooming, dressing, toilet hygiene, functional mobility, and self-feeding. To calculate the ADL, we used a simplified scale which assigned a 0 or 1 to each self-care task according to the level of independence. Thus, the total score can range between 0 and 6. 

### 2.3. Statistical Analysis

Continuous data were presented as means and standard deviations, while categorical data were presented as percentages and counts. Differences between groups were calculated using *t*-test or chi-squared, as appropriate. Multiple linear regression was computed to evaluate factors associated with the levels of individual distress only in the group of caregivers of autistic people. Specifically, GHQ-12 scores were inserted as the dependent variable, while age and gender of caregivers, BRCS scores, age and gender of the family member, associated ID, daily caregiving time (hours), and total caregiving period (years) were included as independent variables. Statistical analyses were calculated using SPSS 23.0 for Windows. 

## 3. Results

### 3.1. Characteristics of Participants 

In the present analysis, we included 383 participants, of which 141 declared to be primary caregivers of an autistic person and 242 were primary caregivers of people with other neurodevelopmental or psychiatric disorders (which from now on will be mentioned as “caregivers of non-autistic people”). The sample of caregivers of autistic and non-autistic people significantly differed in mean age (45.53 ± 8.89 in the former vs. 51.84 ± 11.87 in the latter), but not in gender, as the vast majority of them were females (93.6% vs. 90.1%). The two groups did not significantly differ according to geographic origin and educational levels (see Table 1). Conversely, the two groups significantly differed in terms of occupational levels. Particularly, only 5.7% of caregivers of autistic people were retired, in contrast with the 17.8% of the other group of caregivers. Overall, 7% of the sample reported suffering from a psychiatric disorder, with a higher proportion among caregivers of non-autistic individuals (8.7% of caregivers of non-autistic people vs. 4.3% of caregivers of autistic people). There was also a significant difference in the number of cohabitants between the two groups, with a higher percentage of caregivers living only with the family member they were caring for (18.2% of caregivers of non-autistic people vs. 7.8% of caregivers of autistic individuals). Characteristics of the two groups are reported in Table 1. 

### 3.2. Characteristics of Autistic and Non-Autistic Family Members of Study Participants

Primary caregivers were also asked to complete specific questions regarding the person they were taking care of and the duration of care. Both caregivers of autistic and non-autistic people were mainly parents, with a higher prevalence among the former group (97.9% vs. 79.8%). A minor proportion was composed of siblings (2.1% vs. 8.7%), while there were no spouses or children among the caregivers of autistic people. Assisted persons were mainly males in both groups (81.6% vs. 65.3%). Regarding age, the majority of autistic people were children (69.5%), while non-autistic individuals were mainly adults (18–64 years) (54.5%). To estimate the severity of illness we calculated the activities of daily living (ADL, range 0–6): we found significant differences, as the mean ADL score was significantly lower in the group of autistic people than non-autistic individuals.

The majority of caregivers responded they had been taking care of their family members for more than 10 years (48.9% of caregivers of autistic people vs. 55.8% of caregivers of non-autistic people) and more than 10 h per day during the COVID-19 pandemic, with a higher and statistically significant percentage among caregivers of autistic people (67.4% vs. 46.7%). Nearly half of participants reported that their efforts had slightly increased during the lockdown, while they were increased for almost 30% of both groups (30.5% vs. 26%). In total, 17% of caregivers of autistic people and 22.7% of other caregivers stated that the fatigue in caregiving was unchanged, while only a small proportion of caregivers (2.1% vs. 2.9%) reported that their efforts were decreased. Specific characteristics of caregivers and their family members are reported in Table 2.

### 3.3. Differences in Psychological Variables between Caregivers of Autistic and Non-Autistic People

Our results showed no significant differences in the examined psychological variables between caregivers of autistic and non-autistic people during the lockdown. Mean scores and standard deviations in the psychological variables evaluated during the online survey are reported in Figure 1.

Considering the cutoff of 12/13, 98.2% of all caregivers (particularly, 97.2% of caregivers of autistic people and 98.8% of caregivers of non-autistic people) showed symptoms of psychological distress during the lockdown period. The two groups did not significantly differ regarding mean scores obtained from the GHQ-12 (21.83 vs. 21.83).

No significant differences were detected in the levels of insomnia. Symptoms of insomnia were present in 66.8% of caregivers. Specifically, 22% of caregivers of autistic people reported clinically significant insomnia vs. 27.3% of other caregivers.

Levels of family distress did not significantly differ according to the FDI. Using the threshold of 12, 23.4% of the caregivers of autistic people and 24% of other caregivers presented high levels of family distress during the lockdown period.

According to the scores obtained at the BRCS, our findings did not show any statistically significant difference between the two groups. Specifically, 39% of caregivers of autistic people and 38.4% of caregivers of non-autistic people were classified as low resilient copers. Overall, 36.9% of caregivers of autistic individuals and 34.4% of other caregivers were classified as medium resilient copers. Finally, 24.1% of caregivers of autistic people were classified as high resilient copers vs. 27.3% of the other group of caregivers. 

### 3.4. Predictors of Individual Distress (GHQ-12) among Autistic Caregivers during the Lockdown Period 

A multiple linear regression was computed to calculate potential predicting variables of the psychological distress experienced by caregivers of autistic people during the lockdown. A significant regression equation was found (F = 3.77, *p* < 0.001), with an adjusted R^2^ of 0.24. We found that the risk of individual distress during the COVID-19 pandemic was higher in people caring for children compared to adults (β = 0.36; *p* = 0.002). Moreover, we found that lower levels of resilience as reported at the BRCS predicted higher levels of individual distress during lockdown among the caregivers of autistic people (β = −0.31; *p* < 0.001). Among the other variables considered in the model, no other independent predictors were found. Results are reported in Table 3.

## 4. Discussion

The present study primarily aimed to evaluate the levels of individual wellbeing, family distress, insomnia, and resilience in caregivers of autistic individuals during the COVID-19 lockdown and compare them with caregivers of people with other neurodevelopmental, psychiatric, or relational disabilities. No significant differences emerged between the two groups of caregivers considered in the present analysis. We could hypothesize that the pandemic has represented such a great stressor that both groups were equally affected. Indeed, in our previous study, we showed that caregivers of people with different types of disabilities reported overall more severe psychological consequences than the general population during the lockdown [31]. Another explanation could be related to the fact that the decreased frequency of social contacts has been positively accepted by a part of the autistic population, which often displays a different style of social communication and makes continuous efforts to camouflage their traits. The relaxation in in-person interactions may have been followed by reduced social demands with a consequent decrease in social anxiety. In turn, this may have counterbalanced the distress caused by alterations in daily routine with less negative consequences on caregiver burden. This is in line with personal accounts of autistic adults who reported that the confinement period has positively impacted their wellbeing due to the lower perceived social pressure and the possibility to behave more spontaneously [29]. The lockdown has been also perceived as an opportunity to spend more time with family members and pets as well as cultivate personal interests without feeling the pressure of work and school environments and their intense schedules [28]. Positive emotions have been also expressed by caregivers of autistic children with a hard time at school. Eventually, self-isolation and social distancing have reduced anxiety levels in these children, thus creating a more relaxed environment for the whole family [42]. 

Our secondary aim was to evaluate factors associated with the psychological distress experienced by caregivers of autistic people. Our data revealed that individual distress was higher in people caring for autistic children compared to autistic adults. This finding is in contrast with a study conducted in Spain, which found no differences in the stress perceived by caregivers of autistic children and adults after the lockdown onset [30]. It has been reported that strategies of caregivers in managing the needs of the autistic family member improve over time [43,44]; therefore, caregivers of adults may be more confident in handling challenging situations. At the same time, autistic adults may have developed more adaptive abilities.

Resilience represented another factor associated with the level of individual distress perceived by caregivers of autistic people. Specifically, we confirmed that lower resilience levels predicted higher levels of individual distress during the lockdown, as reported in our previous study [31], as well as in other studies conducted among the general population [45,46]. Our results are consistent with past literature reporting that higher levels of resilience are associated with lower levels of distress in family caregivers [47]. Resilience can be defined as the capacity for successful adaptation in front of adversities and can be determined by either individual, family, or community factors [48]. In families of autistic individuals, resilience has been associated with social support, coping style, cognitive appraisal, optimism, locus of control, self-efficacy, acceptance, sense of coherence, and positive family functioning [49,50]. Promoting resilient attitudes and positive thinking in autistic people and their caregivers thus appears fundamental [51] and beneficial for both the caregivers and the autistic individuals [52]. In this regard, the COVID-19 pandemic may represent a unique opportunity to foster resilience not only for specific individuals or families but the overall system [53]. Autistic communities have campaigned for improving the way of working, learning, and accessing services to people with disabilities [28]. During the COVID-19 outbreak, telehealth and distance learning have been rapidly implemented. Technological innovations may in the future facilitate the creation of tailored services to meet the needs of autistic individuals and their families [53].

To our knowledge, this is the first study to compare the psychological impact of the lockdown on caregivers of autistic individuals to caregivers of non-autistic people in Italy. We evaluated different outcomes, such as personal wellbeing, family distress, insomnia, and resilience. Nevertheless, some limitations should be acknowledged while discussing the results of the present study. First, the sample size is limited and might not be representative of the entire population of caregivers of people with autism or mental disorders. Indeed, we did not perform a random sampling procedure for participants’ selection. Participants’ family members with a disability differed in terms of socio-demographic characteristics, with a potential different impact on the outcome. Of note, we might have excluded a population of caregivers who have less access to technology. Second, we could not compare the reported scores of the evaluated psychological variables with pre-pandemic levels due to the cross-sectional design. Third, although all scales showed good internal consistency in our sample, the FDI and the BRCS have not been formally validated in the Italian language. Fourth, the tools employed in our study are limited and might not have been sufficiently sensitive to capture the complexity of caregivers’ perceptions during the lockdown. Eventually, a qualitative approach with a collection of individual experiences may allow for exploring caregivers’ perspectives in a more nuanced way.

## 5. Conclusions

In conclusion, our study did not confirm that the lockdown has caused more severe psychological consequences among caregivers of autistic people than other caregivers. The ongoing COVID-19 emergency represents a challenging period for the global population, particularly for caregivers of people with disabilities, such as autistic individuals. The pandemic should be taken as a unique opportunity to explore the perspectives of family caregivers and respond to the needs of people with neurodevelopmental, relational, and psychiatric conditions. Future research should investigate the long-term psychological impact of the COVID-19 outbreak, including the end of the period of confinement, as well as the emergent mental health issues among caregivers, a category at high risk for the development of psychopathological conditions.

## Figures and Tables

**Figure 1 brainsci-12-00116-f001:**
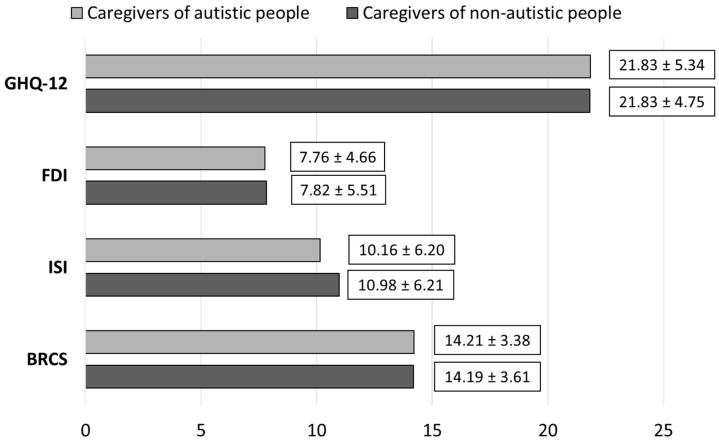
Mean scores ± standard deviations in psychological wellbeing (GHQ-12), family distress (FDI), insomnia (ISI), and resilience (BRCS) reported by caregivers of autistic and non-autistic people. Legend: BRCS: Brief Resilience Coping Scale; FDI: Family Distress Index; GHQ-12: General Health Questionnaire-12 items; ISI: Insomnia Severity Index.

**Table 1 brainsci-12-00116-t001:** Characteristics of caregivers of autistic and non-autistic people.

	Caregivers of Autistic People (N = 141)	Caregivers of Non-Autistic People (N = 242)	t/χ^2^	*p*-Value
Age, mean ± SD	45.53 ± 8.89	51.84 ± 11.87	−5.48	<0.001 *
Gender, female (%)	132 (93.62)	218 (90.08)	1.41	0.23
Region, n (%)			0.07	0.97
Northern Italy	75 (53.19)	128 (52.89)		
Central Italy	25 (17.73)	41 (16.94)		
Southern Italy	41 (29.08)	73 (30.16)		
Education, n (%)			1.55	0.67
Primary or middle school	24 (17.02)	34 (14.05)		
High school	72 (51.06)	117 (48.35)		
Graduate or post-graduate	45 (31.91)	91 (37.60)		
Occupation, n (%)			13.95	0.003 *
Employed	94 (66.67)	146 (60.33)		
Unemployed	39 (27.66)	50 (20.66)		
Retired	8 (5.67)	43 (17.77)		
Student	0 (0)	3 (1.24)		
N of current household members, n (%)			11.58	0.009 *
2	11 (7.80)	44 (18.18)		
3	47 (33.33)	92 (38.02)		
4	60 (42.55)	80 (33.06)		
5 or more	23 (16.31)	26 (10.74)		
Psychiatric disorder, n (%)	6 (4.25)	21 (8.68)	2.66	0.10

Legend: SD = standard deviation; * *p* < 0.05.

**Table 2 brainsci-12-00116-t002:** Characteristics of the autistic and non-autistic family members of the web-survey participants.

	Autistic People (N = 141)	Non-Autistic People (N = 242)	t/χ2	*p*-Value
Age of family member, n (%)			37.31	<0.001 *
Children (<18 yo)	98 (69.50)	94 (38.84)		
Adults (18–64 yo)	43 (30.50)	132 (54.54)		
Older Adults (≥65 yo)	0 (0)	16 (6.61)		
Gender of family member, n (%)			11.52	0.001 *
Female	26 (18.44)	84 (34.71)		
Male	115 (81.56)	158 (65.29)		
Relationship with family member, n (%)			25.80	<0.001 *
Parent	138 (97.87)	193 (79.75)		
Child	0 (0)	18 (7.44)		
Spouse	0 (0)	9 (3.72)		
Sibling	3 (2.13)	21 (8.68)		
ADL, mean ± SD	3.35 ± 1.84	4.32 ± 2.15	4.48	<0.001 *
Duration of caregiving, n (%)			4.31	0.36
Less than 1 year	6 (4.25)	10 (4.13)		
1 to 3 years	12 (8.51)	21 (8.68)		
3 to 5 years	20 (14.18)	19 (7.85)		
5 to 10 years	34 (24.11)	57 (23.55)		
More than 10 years	69 (48.93)	135 (55.78)		
Daily duration of caregiving, n (%)			22.90	<0.001 *
Less than 3 h	5 (3.55)	39 (16.12)		
3 to 5 h	9 (6.38)	31 (12.81)		
5 to 10 h	32 (22.69)	59 (24.38)		
More than 10 h	95 (67.38)	113 (46.69)		
Variation of efforts, n (%)			4.19	0.38
Much increased	43 (30.50)	63 (26.03)		
Slightly increased	71 (50.35)	114 (47.11)		
Unchanged	24 (17.02)	55 (22.73)		
Slightly decreased	3 (2.13)	7 (2.89)		
Much decreased	0 (0)	3 (1.24)		

Legend: ADL = Activities of Daily Living; SD = standard deviation; yo = years old; * *p* < 0.05.

**Table 3 brainsci-12-00116-t003:** Multivariate linear regression evaluating predictors of individual distress (GHQ-12) in the group of autistic caregivers.

Variables	B (95% CI)	Beta	t	*p*-Value
Gender	2.09 (−1.40, 5.58)	0.10	1.19	0.24
Age	0.13 (−0.007, 0.26)	0.21	1.87	0.06
Resilience (BRCS)	−0.49 (−0.73, −0.25)	−0.31	−4.00	<0.001 *
Gender of family member (reference: Male)	−0.67 (−2.85, 1.50)	−0.05	−0.61	0.54
Age of family member (reference: Adult)	4.13 (1.56, 6.69)	0.36	3.18	0.002 *
Associated intellectual disability	−1.98 (−4.30, 0.34)	−0.13	−1.69	0.09
Activities of Daily Living (ADL)	−0.17 (0–66, 0.32)	−0.06	−0.67	0.50
Total duration of caregiving (reference: More than 10 years)				
5 to 10 years	−0.02 (−2.44, 2.40)	−0.002	−0.02	0.99
Less than five years	−0.37 (−2.86, 2.11)	−0.03	−0.29	0.77
Daily hours of caregiving (reference: More than 10 h)				
5 to 10 h	0.14 (−1.95, 2.22)	0.01	0.13	0.90
Less than 5 h	−2.09 (−5.17, 0.98)	−0.12	−1.35	0.18

Legend: BRCS: Brief Resilient Coping Scale; CI: confidence interval; * *p* < 0.05.

## Data Availability

The data are not publicly available.

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
