# Peer review of "The Psychological Impact of the COVID-19 Lockdown: A Comparison between Caregivers of Autistic and Non-Autistic Individuals in Italy"

_brainsci, 2022, doi:10.3390/brainsci12010116_

Round 1

Reviewer 1 Report

Dear authors, thank you for the submission. Here are my review & comments (suggested addition in bold):

  • L32: especially in case of associated intellectual disability (ID) (meaningful reference: Guinchat et al., 2020)
  • L36: Medical comorbidities, such as pain etiologies, epilepsy or metabolic disorders, are also highly prevalent among the ASD population (7-9). Accessibility to specialized healthcare for medical investigations and developmental services for skills improvement, autonomy support and quality of life is therefore needed (suggested references: Guinchat et al., 2020; Nollace et al., 2020).
  • L72: such as age of the subject with ASD, ASD severity, and type of family structure with their daily living habits.
  • L83: “familial” instead of “familiar”
  • L94: In the Materials and Methods section, Study design:

Can you precise the chosen distribution channels? « through social networks »: do you mean family & patient dedicated networks (eg Rare Disease Association) or general networks (eg Twitter/Facebook)?

  • L105: … and the Family Distress Index (FDI, 29). Can you add a sentence about ADL?: Level of independence of the assisted subjects was assessed using the Activities of Daily Living (ADL) (reference).
  • L229: during the lockdown period. L229: “regarding” instead of “regards”.
  • L239: “non-ASD caregivers” instead of “not-ASD caregivers”.
  • L251: “among the caregivers” instead of “the among caregivers”.
  • L255: Table 3. “predictors” instead of “predictor”.
  • P 8/11, Concernant les limitations de l’étude:
  • L307: First, the sample size is limited and might not be representative of the entire population of caregivers of people with mental disorders. (I suggest adding the following): Indeed, we did not realized a random procedure during the participant’s selection procedure.
  • Suggestion: This study focused on the lockdown period and did not include the deconfinement. The deconfinement period, with the resuming of the everyday social activities after 2 months of lockdown could be another interesting transition period with which to compare the reported results and their potential impact on caregivers.
  • L318: Future research should investigate the long-term psychological impact of COVID-19 outbreak, including the end of the period of confinement, as well as….

References suggested (see attached file):

  • Guinchat V, Cravero C, Lefèvre-Utile J, Cohen D. Multidisciplinary treatment plan for challenging behaviors in neurodevelopmental disorders. In A. Gallagher, C. Bulteau, D. Cohen, & J. Michaud (Eds.) Handb Clin Neurol, 2020, 174(22):301-21. doi: 10.1016/B978-0-444-64148-9.00022-3
  • Nollace L, Cravero C, Abbou A, Mazda-Walter B, Bleibtreu A, Pereirra N, Sainte-Marie M, Cohen D, Giannitelli M. Autism and COVID-19: a case series in a neurodevelopmental unit. J Clin Med, 2020;9, 2937. doi:10.3390/jcm9092937

Author Response

Q1. Dear authors, thank you for the submission. Here are my review & comments (suggested addition in bold).

R1. Thank you for accepting to review our manuscript and for the insightful comment. We have tried to do our best to address all the issues raised.

Q2. L32: especially in case of associated intellectual disability (ID) (meaningful reference: Guinchat et al., 2020)

R2. Thank you for suggesting this interesting paper. The reference has been added according with your suggestion.

Q3. L36: Medical comorbidities, such as pain etiologies, epilepsy or metabolic disorders, are also highly prevalent among the ASD population (7-9). Accessibility to specialized healthcare for medical investigations and developmental services for skills improvement, autonomy support and quality of life is therefore needed (suggested references: Guinchat et al., 2020; Nollace et al., 2020).

R3. Thank you very much for your comment. We have modified the text as suggested:

“Medical morbidities, such as epilepsy, metabolic disorders, or pain-related conditions, are also highly prevalent among the autistic population [10-12]. Accessibility to specialized healthcare for medical investigations and developmental services for skills improvement, autonomy support, and quality of life is therefore needed [6,13].”

Q4. L72: such as age of the subject with ASD, ASD severity, and type of family structure with their daily living habits.

R4. Thank you very much for this suggestion. We have modified the sentence as follows:

However, findings regarding the psychological distress of the autistic family members were contradictory, depending on variables such as the age of the autistic subject, the severity of the condition, as well as the type of family structure with their daily living habits [25].”

Q5. L83: “familial” instead of “familiar”

Q5. Thank you. Corrected as per the suggestion.

Q6. L94: In the Materials and Methods section, Study design: Can you precise the chosen distribution channels? « through social networks »: do you mean family & patient dedicated networks (eg Rare Disease Association) or general networks (eg Twitter/Facebook)?

Q6. Thank you for the comment. We have clarified this point as follows:

“A web-based questionnaire was developed by the research team and spread through social networks (i.e., Facebook) between April 19 and May 3, 2020. Moreover, associa-tions dedicated to the support of people with mental and relational disorders and their caregivers were contacted to help spread the questionnaire (see Acknowledgments Section).”

Q7. L105: … and the Family Distress Index (FDI, 29). Can you add a sentence about ADL?: Level of independence of the assisted subjects was assessed using the Activities of Daily Living (ADL) (reference).

R7. Thank you for this important suggestion. We have added a sentence as follows:

The Activities of Daily Living (ADL) scale [38] was used to evaluate the level of independence of autistic and non-autistic family members.”

Q8. L229: during the lockdown period.

R8. Thank you for noticing this missing part of the sentence.

Q9. L229: “regarding” instead of “regards”.

R9. Corrected as per the suggestion.

Q10. L239: “non-ASD caregivers” instead of “not-ASD caregivers”.

R10. Thank you. Corrected as per the suggestion.

Q11. L251: “among the caregivers” instead of “the among caregivers”.

R11. Corrected as per the suggestion.

Q12. L255: Table 3. “predictors” instead of “predictor”.

R12. Corrected. Thank you.

Q13. P 8/11, Concernant les limitations de l’étude: L307: First, the sample size is limited and might not be representative of the entire population of caregivers of people with mental disorders. (I suggest adding the following): Indeed, we did not realized a random procedure during the participant’s selection procedure.

R13. Thank you for this insightful consideration. We have added the suggested sentence.

Indeed, we did not perform a random sampling procedure for participants’ selection.”

Q14. Suggestion: This study focused on the lockdown period and did not include the deconfinement. The deconfinement period, with the resuming of the everyday social activities after 2 months of lockdown could be another interesting transition period with which to compare the reported results and their potential impact on caregivers. Q17. L318: Future research should investigate the long-term psychological impact of COVID-19 outbreak, including the end of the period of confinement, as well as….

R14. Thank you very much. We have added this important consideration in the Conclusions, as suggested by the Reviewer.

Reviewer 2 Report

The topic of this paper is of relevance and importance in the aftermath of the COVID health restrictions imposed on supports and services for autistic people and their caregivers.

I think the authors could reconsider the title as the paper does not only relate to psychological distress, but a range of indicators of caregiver well being and this is not reflected in the current title. The use of the term autistic people is welcome in the title.

I am concerned about the language and terminology used throughout the paper that reverts to medical model framing of autism which is not in keeping with current approaches. The use of the term autism spectrum disorder is not welcomed by many segments of the autism community and could be avoided in this paper. The use of identity first language, and removal of chronic disease, patients, illness etc is required to meet current expectations. The framing of the participant groups would best be described as caregivers of autistic people and caregivers of non-autistic people which would avoid the use of the ASD acronym.

There are also some areas where further editing is required to align with English grammatical conventions.

The introduction to the paper covers some important concepts and literature. However, I am concerned by the lack of balance in the introduction which is predominantly negative and portraying autistic individuals in a deficit perspective. This needs to be addressed to include more balance and a curiosity about the positive experiences of autistic people and their carers, not just a list of problems.  

The methods are clearly described and appropriate for this study. The description of the data collection tools is clear and measured.

The results are clearly presented and the use of tables is helpful. The use of dated terms is also an issue in the results section and requires some attention to address this. On page 6 you use the terms caregivers of autistic and non-autistic people which is a much more acceptable term and would be helpful to use throughout the paper.

The discussion begins with a summary of the purpose of the paper which is not necessary. It would be more helpful to summarise the findings in the context of the purpose of the paper. The discussion requires further development as you do not explore the findings that you have presented in a comprehensive manner. The concepts around positive outcomes for autistic individuals could be expanded to link more closely with other literature, particularly if you revise your introduction to include a more balanced view of the strengths of autistic individuals.

You could expand on the factors associated with caregiver well being and explore this concept further in relation to COVID related issues. The current paragraph does not link your findings with the COVID issues which is a missed opportunity as there is current literature you could draw on to explore these factors. The resilience paragraph requires more development as you do make a link with a relevant article, but do not develop the ideas or present the similarities and differences.

Your discussion of limitations also requires further development. A consideration of the relevance and sensitivity of the instruments used in the study is worth discussing as well as the profile of the participant group.  You could also point to the value of obtaining caregiver perspectives in a more individualised manner to supplement standardised instruments to flesh out the experience in a more nuanced way.

This paper has the potential to make a great contribution to the evidence base about autistic individuals and their caregivers experience of the COVID lockdowns, but requires further revision.

Author Response

Q1. The topic of this paper is of relevance and importance in the aftermath of the COVID health restrictions imposed on supports and services for autistic people and their caregivers.

R1. Thank you very much for accepting to revise our paper and for the insightful comments provided.

Q2. I think the authors could reconsider the title as the paper does not only relate to psychological distress, but a range of indicators of caregiver well being and this is not reflected in the current title. The use of the term autistic people is welcome in the title.

R2. Thank you. We have now modified the title as follows:

“The psychological impact of the Covid-19 lockdown: a comparison between caregivers of autistic and non-autistic individuals in Italy”

Q3. I am concerned about the language and terminology used throughout the paper that reverts to medical model framing of autism which is not in keeping with current approaches. The use of the term autism spectrum disorder is not welcomed by many segments of the autism community and could be avoided in this paper. The use of identity first language, and removal of chronic disease, patients, illness etc is required to meet current expectations. The framing of the participant groups would best be described as caregivers of autistic people and caregivers of non-autistic people which would avoid the use of the ASD acronym.

R3. Thank you very much for your insightful comments. In line with part of the autistic community, we have now used identity-first language and avoid the definition of “autism spectrum disorder” throughout the paper. Particularly, we have referred to the guidelines provided by the journal “Autism in adulthood” (https://home.liebertpub.com/publications/autism-in-adulthood/646/for-authors) for the use of specific, non-stigmatizing, terminology. Nevertheless, we have decided to keep the paragraph about the definition of autism according to the medical model, as well as the description of potential co-occurring conditions that may potentially affect autistic people. In our opinion, this is important to emphasize the careful attention needed by the autistic community as well as the burden experienced by a large part of caregivers of autistic people, especially those with profound autism.

Q4. There are also some areas where further editing is required to align with English grammatical conventions.

R4. We have thoroughly revised the paper with the help of a native English speaker and switched to identity-first language. We hope that grammar has now improved.

Q5. The introduction to the paper covers some important concepts and literature. However, I am concerned by the lack of balance in the introduction which is predominantly negative and portraying autistic individuals in a deficit perspective. This needs to be addressed to include more balance and a curiosity about the positive experiences of autistic people and their carers, not just a list of problems.  

R5. Thank you very much for this comment. We agree with the importance of talking about positive experiences lived by autistic people and their carers during the lockdown. Accordingly, we have substantially modified the Introduction, as reported below.

Autistic people and their families have been diversely impacted by confinement. According to a recent systematic review, parents and caregivers of autistic individuals reported an overall increase in perceived stress and a decrease in psychological well-being during the Covid-19 outbreak [25]. However, findings regarding the psychological distress of the autistic family members were contradictory, depending on variables such as the age of the autistic subject, the severity of the condition, as well as the type of family structure with their daily living habits [25]. Focusing specifically on the Italian situation, Levante et al. showed that families of autistic children reported greater behavioral problems and parental distress than families of typically developing children during the lockdown [26]. An online survey also revealed that managing changes and restrictions during the lockdown has been very challenging for parents of autistic children. The majority of parents reported an increase in the burden of care compared to the pre-pandemic period [27]. Despite the enormous disruptions caused by the lockdown, experiences were not always described as straightforwardly negative in the literature. Some autistic adults reported for instance more flexibility, time, and space to pursue their own interests and passions as well as to stay with their families [28]. Others described that the confinement period has positively impacted their mental health, due to the less perceived societal expectations and the opportunity to behave in a more spontaneous way, without spending too much effort in camouflaging behaviors [29]. A study conducted in Spain showed that autistic adults perceived a decrease in stress levels after the lockdown onset and found an improvement in their psychopathological status after two months of social distancing; conversely, caregivers of both autistic children and adults reported higher stress levels [30].”

Q6. The methods are clearly described and appropriate for this study. The description of the data collection tools is clear and measured.

R6. Thank you very much for the positive comment.

Q7. The results are clearly presented and the use of tables is helpful. The use of dated terms is also an issue in the results section and requires some attention to address this. On page 6 you use the terms caregivers of autistic and non-autistic people which is a much more acceptable term and would be helpful to use throughout the paper.

R7. Thank you very much for your comment. We have now carefully checked the results and switched to a more acceptable and less-stigmatizing language, as reported in R3.

Q8. The discussion begins with a summary of the purpose of the paper which is not necessary. It would be more helpful to summarise the findings in the context of the purpose of the paper.

R8. We have modified the beginning of the Discussion as suggested by the reviewer.

Q9. The discussion requires further development as you do not explore the findings that you have presented in a comprehensive manner. The concepts around positive outcomes for autistic individuals could be expanded to link more closely with other literature, particularly if you revise your introduction to include a more balanced view of the strengths of autistic individuals.

Q10. You could expand on the factors associated with caregiver well being and explore this concept further in relation to COVID related issues. The current paragraph does not link your findings with the COVID issues which is a missed opportunity as there is current literature you could draw on to explore these factors.

R9 and R10. Thank you. We have now revised this part accordingly

“No significant differences emerged between the two groups of caregivers considered in the present analysis. We could hypothesize that the pandemic has represented such a great stressor that both groups were equally affected. Indeed, in our previous study, we showed that caregivers of people with different types of disabilities reported overall more severe psychological consequences than the general population during the lock-down [31]. Another explanation could be related to the fact that the decreased frequency of social contacts has been positively accepted by a part of the autistic population, which often displays a different style of social communication and make continuous efforts to camouflage their traits. The relaxation in in-person interactions may have been followed by reduced social demands with a consequent decrease in social anxiety. In turn, this may have counterbalanced the distress caused by alterations in daily routine with less negative consequences on caregiver burden. This is in line with personal accounts of autistic adults who reported that the confinement period has positively impacted their wellbeing due to the lower perceived social pressure and the possibility to behave more spontaneously [29]. The lockdown has been also perceived as an opportunity to spend more time with family members and pets as well as cultivate personal interests, without feeling the pressure of work and school environments and their intense schedules [28]. Positive emotions have been also expressed by caregivers of autistic children with a hard time at school. Eventually, self-isolation and social distancing have reduced anxiety levels in these children, creating thus a more relaxed environment for the whole family [42].

Q11. The resilience paragraph requires more development as you do make a link with a relevant article, but do not develop the ideas or present the similarities and differences.

R11. We have expanded the paragraph about resilience as follows:

“Resilience represented another factor associated with the level of individual distress perceived by caregivers of autistic people. Specifically, we confirmed that lower resilience levels predicted higher levels of individual distress during the lockdown, as reported in our previous study [31], as well as in other studies conducted among the general population [45,46]. Our results are consistent with past literature reporting that higher levels of resilience are associated with lower levels of distress in family caregivers [47]. Resilience can be defined as the capacity for successful adaptation in front of adversities and can be determined by either individual, family, or community factors [48]. In families of autistic individuals, resilience has been associated with social support, coping style, cognitive appraisal, optimism, locus of control, self-efficacy, acceptance, sense of coherence, and positive family functioning [49,50]. Promoting resilient attitudes and positive thinking in autistic people and their caregivers appears thus fundamental [51] and beneficial for both the caregivers and the autistic individuals [52]. In this regard, the Covid-19 pandemic may represent a unique opportunity to foster resilience not only for specific individuals or families but the overall system [53]. Au-tistic communities have campaigned for improving the way of working, learning, and accessing services to people with disabilities [28]. During the Covid-19 outbreak, tele-health and distance learning have been rapidly implemented. Technological innovations may in the future facilitate the creation of tailored services to meet the needs of autistic individuals and their families [53].”

Q12. Your discussion of limitations also requires further development. A consideration of the relevance and sensitivity of the instruments used in the study is worth discussing as well as the profile of the participant group. You could also point to the value of obtaining caregiver perspectives in a more individualised manner to supplement standardised instruments to flesh out the experience in a more nuanced way.

R12. We have now expanded the Limitation section as follows:

First, the sample size is limited and might not be representative of the entire population of caregivers of people with autism or mental disorders. Indeed, we did not perform a random sampling procedure for participants’ selection. Participants’ family members with disability differed in terms of socio-demographic characteristics, with a potential different impact on the outcome. Of note, we might have excluded a part of caregivers having less access to technology.”

Fourth, the tools employed in our study are limited and might not have been sufficiently sensitive to capture the complexity of caregivers’ perceptions during the lockdown. Eventually, a qualitative approach with a collection of individual experiences may allow exploring caregivers’ perspectives in a more nuanced way.”

Q14. This paper has the potential to make a great contribution to the evidence base about autistic individuals and their caregivers experience of the COVID lockdowns, but requires further revision.

R14. Thank you again for your insightful suggestions. We have tried to do our best to address the points raised and hope that the manuscript has substantially improved thanks to your comments.

Reviewer 3 Report

Dear Authors

A very interesting and much needed article in times of pandemic.

I would like to make some suggestions to improve the article.

In the introduction, when talking about the diagnosis of ASD, it would be good to mention the DSM-5 and to talk about the dimensionality of the severity defined in it.

It would also be interesting to know if during the first wave people with autism were allowed to go out in the street. In Spain they were given permission and were stigmatised.

The instruments are very well explained but you should point out their reliability and validity in the Italian language.

A major limitation of the present study is that it is almost a year and a half since it was conducted. It also lacks depth of discussion.

Author Response

Q1. Dear Authors, A very interesting and much needed article in times of pandemic. I would like to make some suggestions to improve the article.

R1. We thank the Reviewer for carefully revising our manuscript and for the meaningful suggestions. 

Q2. In the introduction, when talking about the diagnosis of ASD, it would be good to mention the DSM-5 and to talk about the dimensionality of the severity defined in it.

R2. Thank you. According to the suggestions made by Reviewer #2, we decided to adopt identity-first language throughout the paper, avoiding the use of the expression “autism spectrum disorder”. However, we still kept the presentation of autism according to DSM-5 and mentioned the dimensionality of the severity defined according to the medical model, as reported below:

Autism is a life-long neurodevelopmental condition that, according to the Diag-nostic and Statistical Manual for Mental Disorders, Fifth Edition (DSM-5), is character-ized by difficulties in social communication and the presence of restricted patterns of interests or behaviors [1]. Characteristics of autism are widely variable in their presen-tation and can sometimes be masked due to learned coping or camouflaging strategies [2,3]. For this reason, the medical model of autism has proposed three levels of severity, depending on the support needed by the autistic individual [1].”

Q3. It would also be interesting to know if during the first wave people with autism were allowed to go out in the street. In Spain they were given permission and were stigmatised.

R3. Thank you very much for your question. Indeed, in Italy people with disabilities (including autism) were allowed to stay without protective masks “if not compatible with the type of disability”. Moreover, they were allowed to go out for “outdoor activities” if essential. However, we do not have any specific data about the stigmatization of such a particular situation and therefore any comment on this would be speculative. However, we agree with the importance of underlying this exemption and reported this aspect in the Introduction.

In this regard, the Italian legislation has contemplated some exceptions to the rigid rules imposed on the general population. In fact, individuals with disabilities, including autistic people, were allowed to circulate without protective masks and perform outdoor activities if needed (DPCM 12/03/2020, art.12).”

Q4. The instruments are very well explained but you should point out their reliability and validity in the Italian language.

R4. Thank you for your comment. For main outcome variables (GHQ-12, FDI, ISI, and BRCS) we have calculated the Chronbach’s alpha found in our sample. Moreover, for GHQ-12 and ISI we have the internal consistency as reported by the validation studies.

Unfortunately, the FDI and the BRCS have not been officially validated in Italian. We have added this point among the study limitations:

“Third, although all scales showed good internal consistency in our sample, the FDI and the BRCS have not been formally validated in the Italian language.”

Q5. A major limitation of the present study is that it is almost a year and a half since it was conducted. It also lacks depth of discussion.

R5. We have now integrated the Discussion also in light of the comments made by Reviewer 2.

As far as concerns the fact that the study has been conducted almost one year ago, we agree with the Reviewer, but we still believe in the importance of disseminating the findings as they might be useful for caregivers of autistic (and non-autistic people) to cope with future periods of confinement or similar circumstances. Moreover, we have further underlined the importance of evaluating the long-term impact of lockdown in the studied population in the Conclusion.

Future research should investigate the long-term psychological impact of Covid-19 outbreak, including the end of the period of confinement, as well as the emergent mental health issues among caregivers, a category at high risk for the developmental of psychopathological conditions.”

Round 2

Reviewer 2 Report

I thank the authors for their careful consideration of the feedback provided. I appreciate the detail and comprehensiveness of the response and alterations to the paper. 

Author Response

We thank again the Reviewer for the suggestions made, which have undoubtedly contributed to improve the quality of our manuscript. Your comments have been really appreciated.

Reviewer 3 Report

Dear authors. You have made all the proposed changes. However, you should go more deeply into the requests you are proposing so that the same thing does not happen again in the future. Thank you

Author Response

We would like to thank the Reviewer for the response, although we find the Reviewer's comment a bit unclear. It is really difficult for us to understard which are the "requests" he/she is referring to. In our revised version of the paper, we have discussed the potential reasons for no differences in the psychological variables considered between the groups of caregivers and proposed some clues and opportunities for future developments. We would like to kindly ask the Reviewer/Editor to clarify which points still need to be expanded. After that, we will be glad to further integrate our paper. Thank you very much.